# In Vivo Neuropharmacological Potential of *Gomphandra tetrandra* (Wall.) Sleumer and In-Silico Study against β-Amyloid Precursor Protein

**Md. Saidur Rahman** [1,2], **Md. Nazmul Hasan Zilani** [3], **Md. Aminul Islam** [1,2], **Md. Munaib Hasan** [3], **Md. Muzahidul Islam** [1,2], **Farzana Yasmin** [1,2], **Partha Biswas** [1,2], **Akinori Hirashima** [4], **Md. Ataur Rahman** [5,6,7,*], **Md. Nazmul Hasan** [1,2,*] and **Bonglee Kim** [6,7,*]

1   Department of Genetic Engineering and Biotechnology, Faculty of Biological Science and Technology, Jashore University of Science and Technology, Jashore 7408, Bangladesh; saidurriyad24@gmail.com (M.S.R.); aminuljust1996@gmail.com (M.A.I.); muzahidul@just.edu.bd (M.M.I.); farzana.just@gmail.com (F.Y.); partha_160626@just.edu.bd (P.B.)
2   Laboratory of Pharmaceutical Biotechnology and Bioinformatics, Department of Genetic Engineering and Biotechnology, Jashore University of Science and Technology, Jashore 7408, Bangladesh
3   Department of Pharmacy, Faculty of Biological Science and Technology, Jashore University of Science and Technology, Jashore 7408, Bangladesh; zilani.phar@just.edu.bd (M.N.H.Z.); munaib_151010@student.just.edu.bd (M.M.H.)
4   Laboratory of Pesticide Chemistry, Division of Bio-Science and Biotechnology, Faculty of Agriculture, Kyushu University, Fukuoka 812-8581, Japan; ahirasim@agr.kyushu-u.ac.jp
5   Global Biotechnology & Biomedical Research Network (GBBRN), Department of Biotechnology and Genetic Engineering, Faculty of Biological Sciences, Islamic University, Kushtia 7003, Bangladesh
6   Department of Pathology, College of Korean Medicine, Kyung Hee University, Seoul 02447, Korea
7   Korean Medicine-Based Drug Repositioning Cancer Research Center, College of Korean Medicine, Kyung Hee University, Seoul 02447, Korea
*   Correspondence: ataur1981rahman@hotmail.com (M.A.R.); mn.hasan@just.edu.bd (M.N.H.); bongleekim@khu.ac.kr (B.K.)

**Abstract:** Medicinal plants possess a surplus of novel and biologically active secondary metabolites that are responsible for counteracting diseases. Traditionally, *Gomphandra tetrandra* (Wall.) Sleumer is used to treat mental disorders. The present research was designed to explore phytochemicals from the ethanol leaf extract of *Gomphandra tetrandra* (Wall.) Sleumer to identify the potential pharmacophore(s) in the treatment of neurological disorders. The chemical compounds of the experimental plant were identified through GC-MS analysis. In-vitro antioxidant activity was assessed using different methods. Furthermore, in-vivo neurological activity was assessed in Swiss-albino mice. Computer-aided analysis was appraised to determine the best-fit phytoconstituent of a total of fifteen identified compounds in the experimental plant extract against beta-amyloid precursor protein. The experimental extract revealed fifteen compounds in GC-MS analysis and the highest content was 9, 12, 15-octadecatrienoic acid (z,z,z). The extract showed potent antioxidant activity in in-vitro assays. Furthermore, in in-vivo neurological assays, the extract disclosed significant ($p < 0.05$) neurological activity. The most favorable phytochemicals as neurological agents were selected via ADMET profiling, and molecular docking was studied with beta-amyloid precursor protein. In the computer-aided study, 1, 5-diphenyl-2h-1, 2, 4-triazoline-3-thione (Pub Chem CID: 2802516) was more active than other identified compounds with strong binding affinity to beta-amyloid precursor protein. The present in vivo and in silico studies revealed neuropharmacological features of *G. tetrandra* leaf extract as a natural agent against neurological disorders, especially Alzheimer's disease.

**Keywords:** neurological activity; GC-MS; ADMET profile; molecular docking; beta-amyloid precursor protein; Alzheimer's disease

## 1. Introduction

The World Health Organization (WHO) approximates that more than a billion people suffer from psychological disorders globally. These include bipolar disorder, traumatic disorders, epilepsy, schizophrenia, Alzheimer's disease, Parkinson's disease, brain tumors, neuro infections, and cerebrovascular disorders like stroke and migraine. Among these diseases, Alzheimer's disease (AD) is prominent in older age groups. It is an advanced neurodegenerative brain disorder that causes structural and functional damage to the brain. Clinically, AD is characterized by unconscious behavior, memory impairment, lack of emotion, dysfunctional changes in language and speech, fatigue, hallucinations, lack of self-sufficiency, a decline in muscle mass, and dependency on caretakers [1,2]. Physiologically, AD is caused due to mitochondrial dysfunction, formation of reactive species (oxygen and nitrogen), lipid peroxidation, nitrosative stress, protein aggregation, protein oxidation, amyloidopathy, tauopathies, CREB signaling pathway, GSK-3 hypothesis, DNA damage, depletion of endogenous antioxidant enzymes, proteasome dysfunction, microglial activation, neuroinflammation, neuroepigenetic modification, etc. [1,3–5]. Among these causes, the amyloid hypothesis, neurotransmitter hypothesis, tau propagation hypothesis, and mitochondrial hypothesis were reported to be tested at 22.3%, 19.0%, 12.7%, and 7.9%, respectively, of all clinical trials up to 2019 [6]. It is noteworthy that it is thought that AD is caused by the pathological accumulation of amyloid-peptides leading to the formation of neurofibrillary tangles and loss of neurons. Most of the current drugs used to treat psychological disorders have obtrusive unwanted effects. This is one of the main obstacles to using these drugs for years. In this perspective, medicinal plants can be a light in the dark. The use of medicinal plants to cure these psychological disorders is seen in ancient scholastic works. Therefore, based on folklore use, an integral part of ethnopharmacology, and scientific evaluation, medicinal plants can serve as a reservoir for the invention of novel bioactive pharmacophores as medicinal plants possess a plethora of novel and biologically active secondary metabolites [7]. Additionally, previous studies have reported the traditional use of *Bacopa monnieri* (L.) *Pennell*, *Celastrus paniculatus* Willd., *Centella asiatica* (L.) *Urban*, *Clitoria ternatea* L., *Convolvulus pluricaulis Choisy*, *Curcuma longa* L., *Desmodium gangeticum* (L.) *DC.*, *Evolvulus alsinoides* L., *Ginkgo biloba* L.,*Glycyrrhiza glabra* L., *Melissa officinalis* L., *Moringa oleifera Lam.*, *Salvia officinalis* L., *Tinospora cordifolia (Thunb)Miers*, *Withania somnifera* (L.) *Dunal* etc. in Alzheimer's disease [8–10].

*Gomphandra tetrandra* (Wall.) Sleumer (Family: Stemonuraceae) is a potential ethnomedicinal evergreen forest shrub available in the hilly regions of Bangladesh. Locally, it is known as Sundalli or Kambuli. It is mostly found in South and Southeast Asia, as well as in Indo-Malaysia and Indochina. In folklore medicine, leaves of *G. tetrandra* are used to treat epilepsy, convulsions, mental problems, as a tonic, etc. [11]. An alkaloid, camptothecin, has been reported from *G. tetrandra* [12]. Until now, very few studies have been reported on *G. tetrandra*. Consequently, this study focused on the identification of bioactive phytochemicals, in-vivo evaluation of neuropharmacological potential, and in-silico study of the beta-amyloid precursor protein responsible for Alzheimer's disease.

## 2. Material and Methods

### 2.1. Plant Material Collection and Identification

The leaves of *Gomphandra tetrandra* were hoarded from the natural forest of Rangamati, Chittagong, Bangladesh. The sample was collected after getting permission and under the supervision of the forest department. Then, it was identified (DACB-54910) by the Bangladesh National Herbarium, Dhaka, Bangladesh, and an office sample was consigned there.

### 2.2. Experimental Animals

Young Swiss-albino mice (20–25 g weight and six weeks old) were adapted to $24 \pm 1\ ^\circ\text{C}$ with 50–70% relative humidity, a 12 h light/dark cycle, and fed a typical diet and water accurately. The experimental mice were purchased from the animal research branch of the

International Center for Diarrheal Disease and Research, Bangladesh (ICDDR, B). All the experimental procedures related to the animal model were performed by the European Community Guideline (EEC directives of 1986; 88/609/EEE) and the ethical standard was approved by the Ethical Review Committee, Faculty of Biological Science and Technology, Jashore University of Science and Technology [Ref: ERC/FBS/JUST/2020-47].

### 2.3. Extract Preparation

The accumulated sample was purified from unwanted materials and washed with distilled water prior to shade drying for 2–3 weeks. Then the dried sample was ground into a coarse powder. About 310 g of powder was soaked in 950 mL of ethanol (95%) with episodic agitation and shaking for ten days. The mixture was filtered using Whatman Grade 1 filter paper (Sigma-Aldrich, St. Louis, MO, USA). Then, using a rotary evaporator (RE-100 PRO, DLAB Scientific Inc., Beijing, China) at 50 °C and 40 rpm, the crude extract was obtained and measured to be 6.53 g (2.11% *w/w*).

### 2.4. Total Phenol Content

The total phenol content of the extract was discerned using Folin-Ciocalteau reagent [13]. In total, 5 mL of 10% Folin-Ciocalteau reagent was mixed with the extract solution. Then, a 4 mL $Na_2CO_3$ (7.5% *w/v*) solution was added to the mixture and mixed vigorously by vortexing. Then, the solution was incubated for 30 min at 40 °C in an incubator (Mini Incubator, Digi system, DSI-100D, New Taipei city, Taiwan). Finally, the absorbance was measured at 765 nm (Dynamica Halo DB-20S, Livingston, UK). Gallic acid (500–100 µg/mL) was used as a standard to quantify total phenol content. The total phenol content of the extract was calculated and denoted as mg gallic acid equivalent (GAE) per gram of dry extract.

### 2.5. Total Flavonoid Content

Previously described aluminum chloride colorimetric assay [14] was used to estimate the total flavonoids content of the extract. In total, 100 µL aluminum chloride (1%) and 100 µL potassium acetate (1 M) were mixed with extract solution (1 mg/mL). Then, distilled water (2.7 mL) was added and vortexed properly. Finally, the absorbance was measured at 415 nm (Dynamica Halo DB-20S, Livingston, UK). Quercetin (50–10 µg/mL) was used as a standard to enumerate total flavonoids content. The total flavonoids content was computed and stated as quercetin equivalent (QE) per gram of dry extract.

### 2.6. Gas Chromatography Mass Spectroscopy (GC-MS)

Gas chromatography-mass spectrometry analysis was carried out with a Clarus® 690 gas chromatograph (PerkinElmer, CA, MA, USA) using a column (Elite-35, 30 m × 0.25 mm; PerkinElmer, CA, MA, USA) with 0.25µm film and it was equipped with a Clarus® SQ 8 C mass spectrophotometer (PerkinElmer, CA, MA, USA). A 1µL sample was injected (splitless mode) and pure helium (99.999%) was used as a carrier gas at a constant flow rate (1 mL/min) for a 40 min run time. The sample was analyzed in EI (electron ionization) mode at high energy (70 eV). Though the inlet temperature was constant at 280 °C, column oven temperature was set at 60 °C (for 0 min), raised at 5 °C per minute to 240 °C and held for 4 min. The scan time and mass range were 1 s and 50–600 *m/z*, respectively [15]. The sample compounds were identified compared to the National Institute of Standards and Technology (NIST) database.

### 2.7. DPPH Free Radical Scavenging Assay

The antioxidant potential of the extract was estimated by the DPPH scavenging assay [16]. Three milliliters of DPPH (0.004% *w/v*) were mixed to different concentrations (1024–2 µg/mL) of the extract solution. After an incubation period of 30 min (in a dark place at room temperature), the absorbance was taken at 517 nm (Dynamica Halo DB-20S, Livingston, UK). In this test, ascorbic acid was used as a standard antioxidant for comparison. The DPPH scav-

enging potential was quantified as: Scavenging (%) = $[1 - (A_{Sample/standard}/A_{control})] \times 100$. From this data, the $IC_{50}$ value was calculated and compared to ascorbic acid.

### 2.8. Ferric Reducing Antioxidant Power (FRAP) Assay

The ferric reducing antioxidant power of the experimental extract was assessed by FRAP reagent [17] with slight modifications. The FRAP reagent was freshly prepared by mixing acetate buffer (pH-3.6), TPTZ solution (10 mM), and aluminum chloride solution (20 mM) at a ratio of 10:1:1. Then, 3 mL was mixed with different concentrations (10–50 µg/mL) of extract solution. After 30 min of incubation at 37 °C, the absorbance was measured at 593 nm (Dynamica Halo DB-20S, Livingston, UK). Ascorbic acid was used as a standard. The reducing power of the extract was denoted as milligrams of ascorbic acid equivalent (AAE) per gram of dry extract.

### 2.9. Open Field Test

An in vivo open field test was conducted on mice based on a reported method [18]. An open field test board is a chessboard-like smooth board with a 0.5 $m^2$ area. It consists of small black and white colored squares with a wall in the border. Twenty mice were randomly assigned into 4 groups containing 5 mice in each. Group-I was treated with distilled water orally at 10 mL/kg, group-II received diazepam at 1 mg/kg, and group-III and IV were taken the extract at 250 and 500 mg/kg, respectively. Animals were kept in one of the corners of the open field board after administration of the samples. The number of small squares crossed by the mice was counted for 3 min at 0, 30, 60, 90, and 120 min from the sample administration time. A room with a calm and quiet environment was used to perform the experiment. The percent movement inhibition of the mice by the extract was calculated as

$$\% \text{ of movement inhibition} = \frac{Mc - Mt}{Mc} \times 100$$

where Mc indicates the mean number of movements in the control group and Mt denotes the mean number of movements in the test group.

### 2.10. Hole Board Test

The hole board test was performed as stated by the previously reported method [19]. A hole board apparatus is a smooth and plain board (45 cm × 45 cm) having 16 circular small holes with uniform spacing. In this test, 20 Swiss-albino mice were selected and randomly divided into 4 small groups comprising 5 mice in each group. Group-I and Group-II were considered as the control and the standard groups and received 10 mL/kg and 1 mg/kg, respectively. Distilled water was taken as the control and diazepam was used as the standard in this experiment. Both Group-III and Group-IV were the test groups and used to orally administer 250 and 500 mg/kg of the extract. Each mouse was placed on the hole board apparatus after 30 min of sample administration to record the number of head dips by individual mice for 10 min. The percent movement inhibition of the mice by the extract was calculated as

$$\% \text{ of movement inhibition} = \frac{Mc - Mt}{Mc} \times 100$$

where Mc indicates the mean number of head dips in the control group and Mt denotes the mean number of head dips in the test group.

### 2.11. Brine Shrimp Lethality Bioassay

To determine the toxicity of the extract, a brine shrimp lethality test was performed [20]. Firstly, *Artemina salina* eggs were hatched in an aquarium in simulated sea water. Fluorescent light was applied over the aquarium for 48 h. Ten adult nauplii were added to different concentrations (1000–31.25 µg/mL) of extract using a pasture pipette. The number

of dead nauplii was examined after 24 h of the incubation period under a magnifying glass to assess the $LC_{50}$ value.

*2.12. In-Silico Study*

2.12.1. Development of Phytochemicals Library

A phytochemical library of 15 compounds found via GC-MS analysis of plant extract was created. The names, PubChem CID, molecular weight, and structures of those compounds were developed from the PubChem database.

2.12.2. Protein Preparation

For the present study, the non-mutated tertiary (3D) structure of the targeted protein (PDB ID: 1AAP) was downloaded initially in PDB format from the Protein Data Bank. Then, the protein was opened in BIOVA Discovery Studio Visualizer Tool 16.1.0 and energy was minimized after selecting the chain, removing the non-bounded residues, water molecules, and eliminating the unwanted portion of the protein. Finally, the 3D structure of the minimized protein was retrieved in PDB format for further molecular docking studies.

2.12.3. Active Site Prediction Using CASTp Server and Grid Generation

In the present study, BIOVA Discovery Studio Visualizer Tool 16.1.0 was used to find the binding site of the desired protein. Moreover, binding pockets over the entire protein were identified using CASTp 3.0. Finally, the receptor grid was generated using the PyRx software.

2.12.4. Absorption, Distribution, Metabolism and Excretion (ADME), and Toxicity Test

For developing a molecule as a drug candidate, it is crucial to evaluate the pharmacokinetic properties of phytochemicals, like adsorption, distribution, metabolism, excretion, and toxicity analysis. The Swiss-ADME server was used to estimate the ADME properties of the compounds. Furthermore, of the 15 compounds, only seven that showed the best pharmacokinetic and drug-like properties through Lipinski's rule of violation were selected for molecular docking and further analysis.

2.12.5. Molecular Docking

Seven selected compounds after pharmacophore analysis were subjected to molecular docking. In the present study, the PyRx tool was used for molecular docking, which is based on Auto Dock Vina. Resultant compounds with a binding affinity (kcal/mol) were retrieved and visualized by BIOVA Discovery Studio Visualizer Tool 16.1.0.

*2.13. Statistical Analysis*

All experiments were repeated independently at least three times and results are presented as mean $\pm$ standard deviation (SD). The in vivo data was analyzed by *t*-tests and one-way ANOVA followed by Turkey's test and Dunnett's comparison test using SPSS software version 20.0 (SPSS, Chicago, IL, USA). Differences were considered to be statistically significant when $p < 0.05$.

## 3. Results

*3.1. Total Phenol Content*

The total phenol content of the extract was estimated using a calibration curve ($Y = 0.00522X - 0.00290$; $R^2 = 0.997$) and it was found to be $86.37 \pm 0.73$ mg (GAE)/g of dried extract.

*3.2. Total Flavonoid Content*

Using the standard quercetin calibration curve ($Y = 0.007990X - 0.008300$; $R^2 = 0.995$), the total flavonoid content of the extract was computed as $111.17 \pm 1.635$ mg quercetin equivalent/g of dry extract.

*3.3. GCMS Analysis*

In GC-MS analysis, 15 compounds were identified from the experimental extract. Figure 1 represents the distinct GCMS chromatogram.

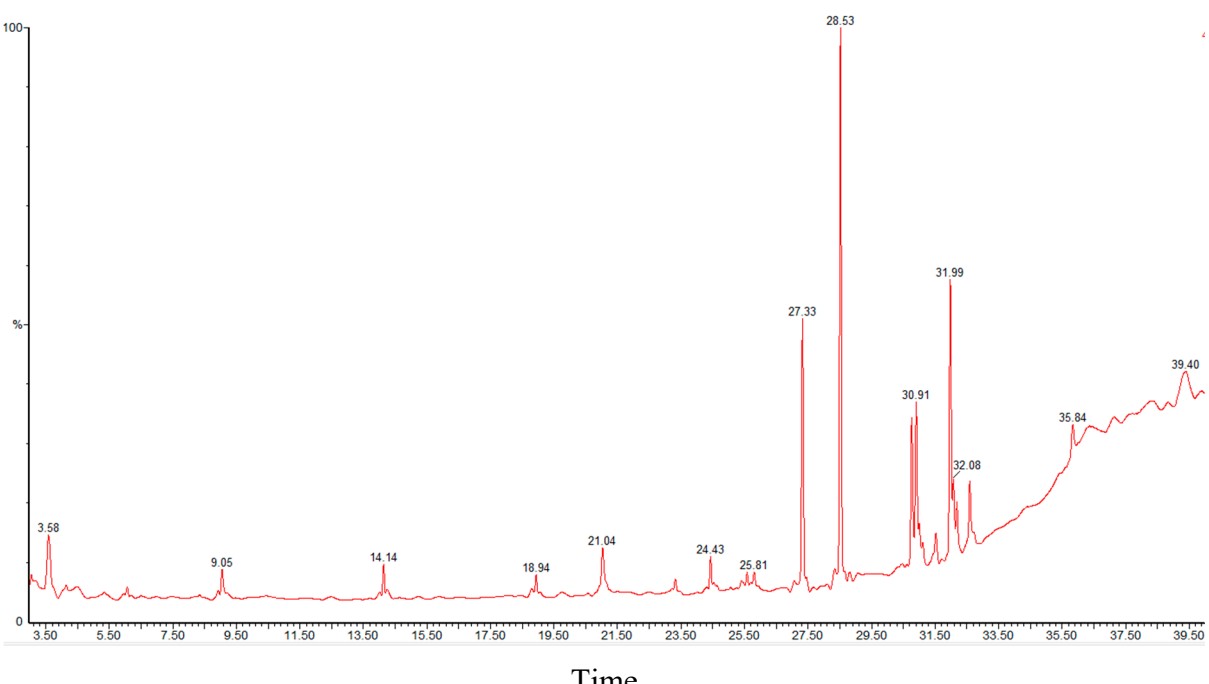

**Figure 1.** Gas Chromatography mass spectroscopic base chromatogram of *G. tetrandra* ethanol leaf extract.

The bioactive compounds were represented by their retention time (RT), molecular formula, molecular weight, and peak area (%) in Table 1. Among 15 compounds, the four major compounds identified are 9,12,15-octadecatrienoic acid, (z,z,z) (37.86%), heptadecanoic acid, ethyl ester (25.897%), 1,2-cyclotadiene (7.625%), and methyl 11-methyl-dodecanoate (6.287%). The other compounds are 6-octadecenoic acid (1.43%), benzene, (1-methylundecyl) (1.26%), o-xylene (0.89%), benzene, 1,3-dimethyl (0.84%), 3-n-hexylthiolane, s,s-dioxide (0.72%), 1-hexanone, 1-phenyl (0.68%), sulfurous acid, nonyl pentyl ester (0.59%), neophytadiene (0.44%), 1,5-diphenyl-2h-1,2,4-triazoline-3-thione (0.41%), chloroacetic acid, tetradecyl ester (0.33%), 13-octadecenoic acid, methyl ester (0.28%) and heptadecanoic acid, ethyl ester (0.13%).

**Table 1.** Gas Chromatography-Mass spectroscopic data of compounds in *G. tetrandra* ethanol leaves extract.

| Serial No. | Retention Time | Name of the Compound | Molecular Formula | Molecular Weight (g/mol) | % Peak Area |
|---|---|---|---|---|---|
| 1 | 3.58 | O-Xylene | $C_8H_{10}$ | 106 | 0.8942 |
| 2 | 4.11 | Benzene, 1,3-dimethyl- | $C_{12}H_{16}$ | 106 | 0.8402 |
| 3 | 5.45 | Benzene, (1-methylundecyl)- | $C_{17}H_{28}$ | 246 | 1.2634 |
| 4 | 6.01 | 1-Hexanone, 1-phenyl- | $C_{12}H_{16}O$ | 176 | 0.6826 |
| 5 | 9.05 | 3-n-Hexylthiolane, s,s-dioxide | $C_{10}H_{20}O_2S$ | 204 | 0.7248 |
| 6 | 18.94 | Chloroacetic acid, tetradecyl ester | $C_{16}H_{31}ClO_2$ | 290 | 0.3253 |
| 7 | 21.04 | Sulfurous acid, nonyl pentyl ester | $C_{14}H_{30}O_3S$ | 278 | 0.5913 |
| 8 | 24.43 | Neophytadiene | $C_{20}H_{38}$ | 278 | 0.4438 |
| 9 | 25.81 | 1,5-Diphenyl-2h-1,2,4-triazoline-3-thione | $C_{14}H_{11}N_3S$ | 253 | 0.4150 |
| 10 | 27.33 | Methyl 11-methyl-dodecanoate | $C_{14}H_{28}O_2$ | 228 | 6.2876 |
| 11 | 28.53 | Heptadecanoic acid, ethyl ester | $C_{19}H_{38}O_2$ | 298 | 25.8970 |
| 12 | 29.83 | 13-Octadecenoic acid, methyl ester | $C_{19}H_{36}O_2$ | 296 | 0.2756 |
| 13 | 30.91 | 1,2-Cyclooctadiene | $C_8H_{12}$ | 108 | 7.6252 |
| 14 | 31.54 | 6-Octadecenoic acid | $C_{18}H_{34}O_2$ | 282 | 1.4305 |
| 15 | 31.99 | 9,12,15-Octadecatrienoic acid, (z,z,z)- | $C_{18}H_{30}O$ | 278 | 37.8608 |

### 3.4. DPPH Free Radical Scavenging Assay

Both the extract and the ascorbic acid showed a concentration-dependent DPPH scavenging activity. The $IC_{50}$ value of the extract was found to be $276.64 \pm 2.91$ µg/mL and the standard was $19.02 \pm 1.26$ µg/mL.

### 3.5. Ferric Reducing Antioxidant Power Assay

The FRAP assay evaluates the antioxidant activity based on the reduction of ferric ($Fe^{3+}$) to ferrous ($Fe^{2+}$). The FRAP value was calculated using the linear equation ($y = 0.027x - 0.080$, $R^2 = 0.993$) obtained from the ascorbic acid standard curve. In this study, ferric reducing capacity was estimated at $90.07 \pm 0.973$ mg/g of ascorbic acid equivalent.

### 3.6. Open Field Test

The extract exhibited a depletion in the movements of mice which was statistically significant ($p < 0.05$) compared to the control. The movement reduction was incessant from the first observation (30 min) to the final (at 120 min) for both doses (250 and 500 mg/kg) of the extract. At the final observation time, the movement inhibition by the extract at 250 mg/kg was 58.33% and it was higher than that of standard (42.66%) at that particular time. However, at 500 mg/kg dose, the movement inhibition was better than the standard at 90 min (50.86%) and 120 min (69.04%) (Figure 2).

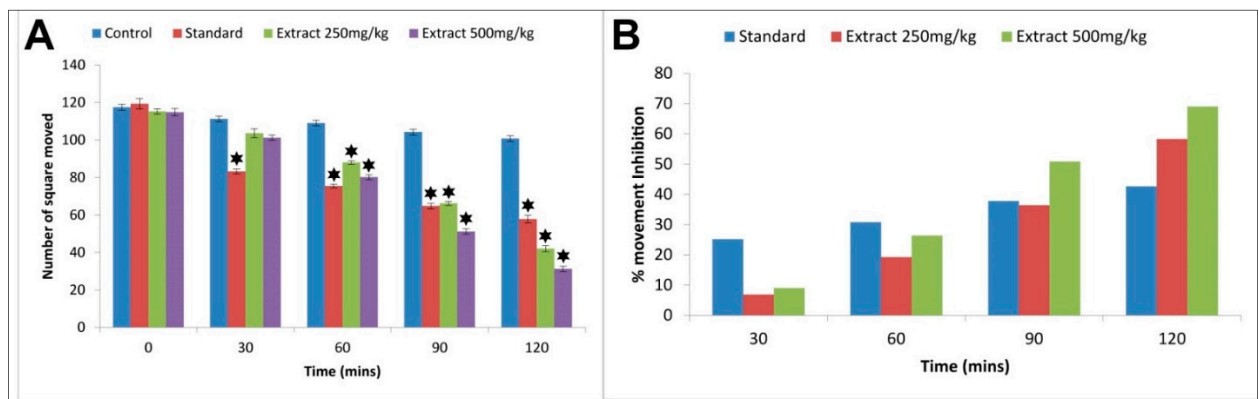

**Figure 2.** Neuropharmacological activity of *G. terandra* in open field test. (**A**) Number of squares moved by mice vs. time, (**B**) percent movement inhibition of mice with time. Here, values are presented as mean $\pm$ SD ($n = 5$). * indicates statistical significance when $p < 0.05$.

### 3.7. Hole Board Test

The extract exhibited a significant ($p < 0.05$) abatement in the number of head dips in mice compared to the effect of the control. The percentage of head dipping inhibition was 16.23% and 28.04% at 250 and 500 mg/kg doses of the extract, respectively, while diazepam at 1 mg/kg showed inhibition of 26.93% (Figure 3). The extract at 250 mg/kg body weight showed better head dipping inhibition than the standard.

### 3.8. Brine Shrimp Lethality Bioassay

The estimated $LC_{50}$ value of the extract was $387.44 \pm 1.46$ µg/mL and for the standard, it was $0.91 \pm 0.74$ µg/mL. The result showed that the extract exhibited little or no cytotoxicity on shrimp larvae.

### 3.9. In Silico Study

3.9.1. Evaluation of Pharmacokinetic Properties

Seven compounds out of the 15 fulfilled all the criteria of the ADMET analysis and showed drug-likeness characteristics (Table 2). That is why they have been selected for further molecular docking analysis. These molecules exhibit an extensive excretion rate

after metabolism in the body. Besides, they also revealed maximum tolerance doses in a range from 0.303 to 1.173 mg/kg/day (Table 2). Moreover, several toxicological parameters, such as hepatotoxicity and AMES toxicity, were assessed and found to be in an acceptable range.

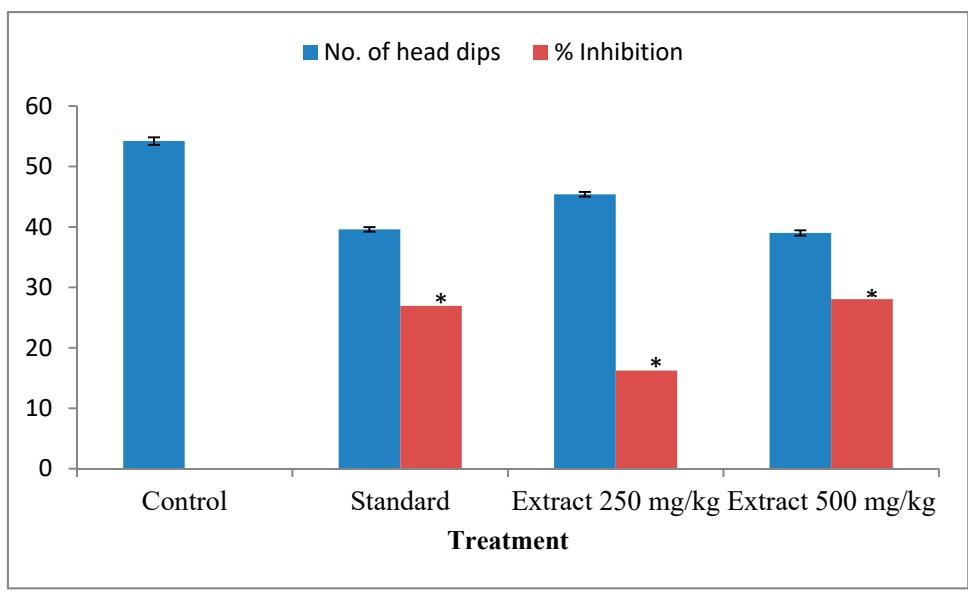

**Figure 3.** Neurological activity of *G. tetrandra* in the hole board test (Values are presented as Mean ± SD) ($n = 5$), * $p < 0.05$ compared to control group.

**Table 2.** The pharmacophore and pharmacokinetic profile of the selected ligand molecules.

| Ligands Name | MW | NHA | NHD | LogP | NRB | GIA | LD50 | BBB | HT | AT | MTD | NLV | DL |
|---|---|---|---|---|---|---|---|---|---|---|---|---|---|
| Rivastigmine (control) | 250.34 | 3 | 0 | 3.21 | 6 | High | 3.402 | Yes | No | No | 0.382 | No | Yes |
| O-Xylene | 106.16 | 0 | 0 | 2.303 | 0 | Low | 1.841 | Yes | No | No | 0.921 | No | Yes |
| 1-Hexanone, 1-phenyl- | 176.25 | 1 | 0 | 3.449 | 5 | High | 1.655 | Yes | No | No | 1.173 | No | Yes |
| 3-n-Hexylthiolane, S,S-dioxide | 204.33 | 2 | 0 | 2.391 | 5 | High | 2.033 | Yes | No | No | 0.393 | No | Yes |
| Sulfurous acid, nonyl pentyl ester | 278.45 | 3 | 0 | 4.539 | 12 | High | 1.98 | Yes | No | No | 0.653 | No | Yes |
| 1,5-Diphenyl-2h-1,2,4-triazoline-3-thione | 253.32 | 1 | 1 | 3.596 | 2 | High | 2.81 | Yes | No | No | 0.926 | No | Yes |
| Methyl 11-methyl-dodecanoate | 228.37 | 2 | 0 | 4.326 | 11 | High | 1.6 | Yes | No | No | 0.303 | No | Yes |
| 1,2-Cyclooctadiene | 108.18 | 0 | 0 | 2.661 | 0 | Low | 2.043 | Yes | No | No | 0.852 | No | Yes |

MW—molecular weight (g/mol); NHA—No. of hydrogen bond acceptor; NHD—No. of hydrogen bond donor; LogP—Predicted octanol/water partition coefficient; NRB—Number of Rotatable Bond; GIA—Gastro-Intestinal absorption (% absorbed); LD50—Oral rat acute toxicity; BBB—Blood-Brain Barrier; HT—Hepatotoxicity; AT—AMES toxicity; MTD—Maximum tolerated dose for humans (log mg/kg/day); NLV—Number of Lipinski's Violation; DL—Drug Likeness.

### 3.9.2. Molecular Docking of the Phytochemicals in the Predicted Ligand-Binding Pocket

Basically, molecular docking is used to understand the biomolecular interactions of the desired compound. Structure-based drug design also takes advantage of this technique. In our study, the top five probable binding pockets in beta-amyloid precursor protein, 1AAP, according to CASTp 3.0 software, were identified (Table 3). The software measured the volume and surface areas (SA) of the desired protein and provided the binding pocket volume and areas (Table 3).

**Table 3.** Predicted top five ligand-binding pockets according to CASTp.

| Serial No. | Pocket ID | Area (Å$^2$) | Volume (Å$^3$) | Pocket Amino Acids |
|:---:|:---:|:---:|:---:|:---:|
| 1 | 1 | 40.093 | 18.554 | TRP 21, TRP 22, CYS 30, PHE 45, TRP 47, GLU 48, GLU 49, CYS 55, GLU 56 |
| 2 | 2 | 32.138 | 16.291 | SER 6, GLU 7, GLN 8, TYR 22, PHE 23, ASP 24, VAL 25 |
| 3 | 3 | 28.618 | 7.281 | ARG 2, CYS 5, SER 6, PHE 23, VAL 25, CYS 55 |
| 4 | 4 | 2.729 | 0.565 | SER 19, PRO 32, PHE 33, PHE 34 |
| 5 | 5 | 4.484 | 0.486 | GLU 10, PHE 33, ASN 41, ASN 43,ASN 44 |

### 3.9.3. Molecular Docking Studies

The beta-amyloid precursor protein (1AAP) has two chains, A and B. A receptor grid with box diameter X = 17.7, Y = − 17.40, Z = 18.54 was prepared for the B chain. The compounds that passed in the ADME and toxicity analysis were studied in the PyRx tool. Among them, only the compound CID2802516 showed the highest binding affinity of −5.5 Kcal/mol. Besides, the compounds CID70337 and CID543842 showed a binding affinity of −4.2 Kcal/mol that was equal to the control ligand CID 77991 (−4.2 kcal/mol) (Table 4).

**Table 4. The** binding affinity of interested ligands with the targeted protein macromolecule and comprehensive intermolecular interactions.

| Ligands Name (PubChem CID) | Binding Affinity (Kcal/mol) | Amino Acid Involved Interaction | |
|:---:|:---:|:---:|:---:|
| | | Hydrogen Bond Interaction | Hydrophobic Bonds Interaction |
| Rivastigmine (77991) | −4.2 | THR47 (3.13 Å); ASP46 (3.01 Å) | CYS55, GLU48, GLU49, and PHE45 |
| O-Xylene (7237) | −4.0 | No H-bond | CYS30, CYS55, GLU48, GLU49, PHE45, and THR47 |
| 1-Hexanone, 1-phenyl- (70337) | −4.2 | No H-bond | ASP46, CYS30, CYS55, GLU48, GLU49, PHE45 and THR47 |
| 3-n-Hexylthiolane,s,s-dioxide (543842) | −4.2 | ASP46 (3.03Å); THR47 (2.86 Å) | CYS30, CYS55, GLU48, GLU49, and PHE45 |
| Sulfurous acid, nonyl pentyl ester (572661) | −3.5 | No H-bond | CYS30, CYS55, GLU48, PHE45 and THR47 |
| 1,5-Diphenyl-2h-1,2,4-triazoline-3-thione (2802516) | −5.5 | GLU49 (3.04 Å) | ASP46, CYS30, CYS55, GLU48, PHE45 and THR47 |
| Methyl 11-methyl-dodecanoate (4065233) | −3.6 | GLY56 (3.34 Å) | CYS30, CYS55, GLU48, PHE45 and THR47 |
| 1,2-Cyclooctadiene (641048) | −3.8 | GLY56 N:O1 (2.91 Å) and N:O2 (3.06 Å) | CYS30, CYS55, GLU48, GLU49, LYS29, and PHE45 |

### 3.9.4. Interpretation of Protein-Ligand Interactions

The Java-based software Ligplot+ V 2.2 was used to identify the hydrogen bond and hydrophobic bond interactions of the protein-ligand complexes. The control drug Rivastigmine (CID: 77991) showed two hydrogen bonds (THR47 (3.13 Å); ASP46 (3.01 Å)) and four hydrophobic bonds (CYS55, GLU48, GLU49, and PHE45) with the desired protein (Figure 5). Among the selected phytochemicals, 3-n-hexylthiolane, s,s-dioxide (CID: 543842) and, 1,2-cyclooctadiene (CID: 641048) displayed a maximum of two hydrogen bonds and more than four hydrophobic bonds, respectively. Each of the compounds 1,5-diphenyl-2h-1,2,4-triazoline-3-thione (CID: 2802516), and methyl 11-methyl-dodecanoate (CID: 4065233) bound with only one hydrogen bond, but the former had six and the latter had five hydrophobic interactions with the protein. Besides, the rest of the compounds did not have any hydrogen bond interactions but had some hydrophobic bonds (Table 4).

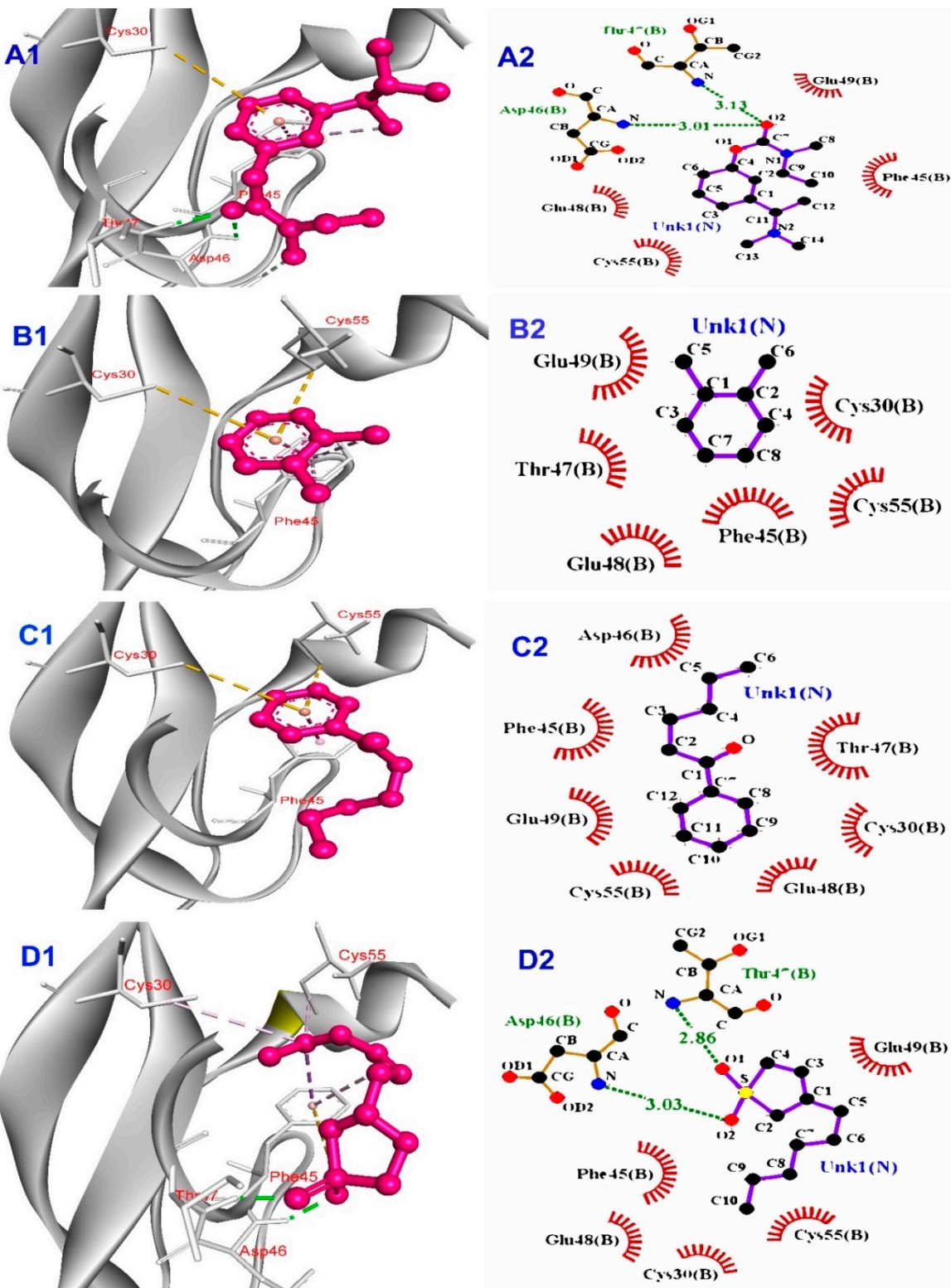

**Figure 4.** *Cont.*

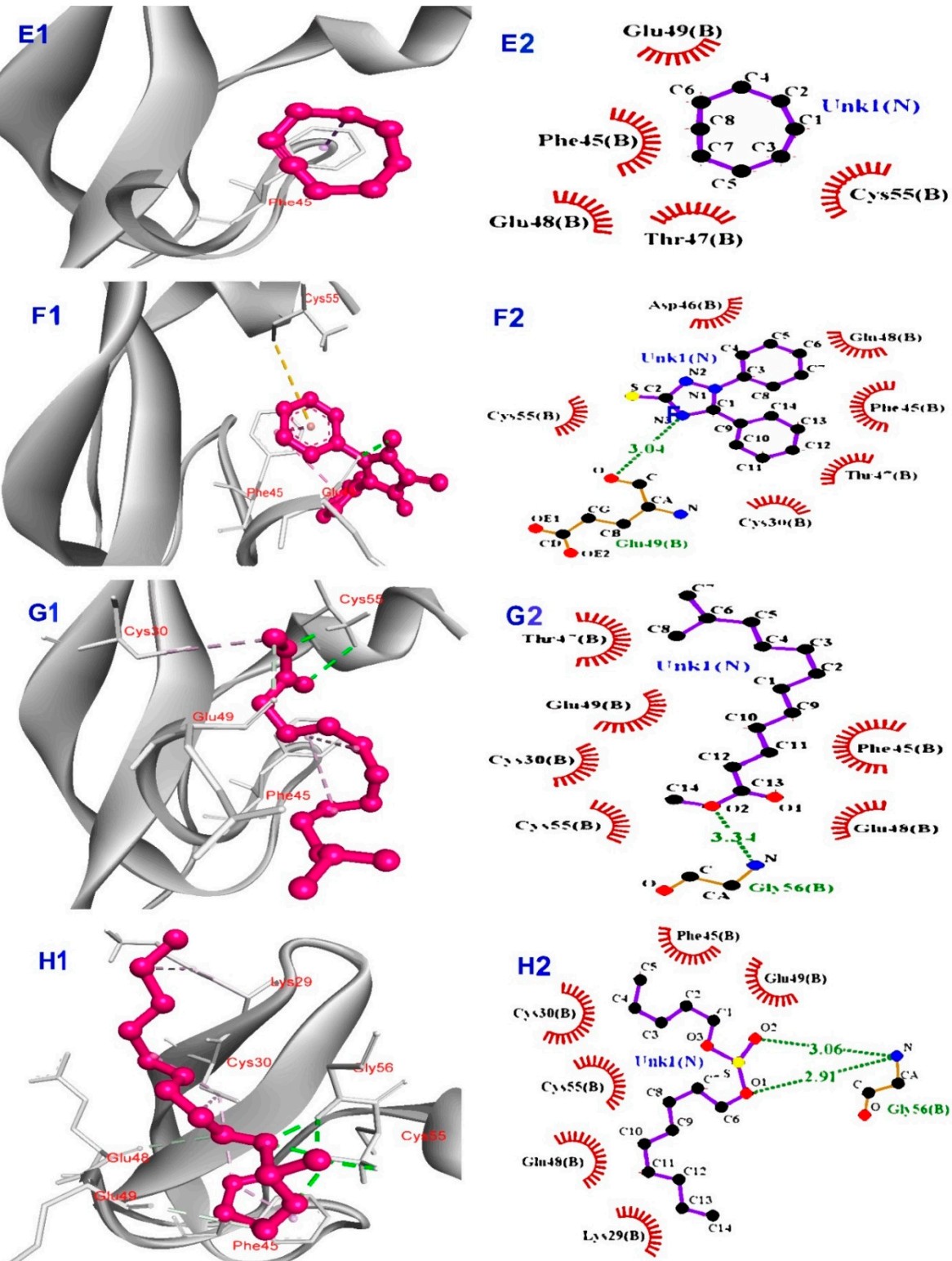

**Figure 5.** Binding of phytochemicals with the targeted protein (1AAP). Here, the left side indicates a 3D structure, and the right side indicates a 2D structure of ligand-beta-amyloid precursor protein-binding complexes. In the 3D structure, ligand molecules were represented in pink color with a ball and stick model and binding amino acids by letters and a number format which are red in color. In the 2D structure, hydrogen bonding interactions are shown by the olive dotted green line and hydrophobic interactions with the nearest amino acid residues in the red spike. (**A**) Control drug Rivastigmine; (**B**) O-Xylene; (**C**) 1-Hexanone, 1-phenyl; (**D**) 3-n-Hexylthiolane, s,s-dioxide; (**E**) Sulfurous acid, nonyl pentyl ester; (**F**) 1,5-Diphenyl-2h-1,2,4-triazoline-3-thione; (**G**) Methyl 11-methyl-dodecanoate; (**H**) 1,2-Cyclooctadiene.

## 4. Discussion

Ethno-medicinal plants have served as a convenient and efficient source of medicine since ancient times. The popularity of traditional medicine is growing at a startling pace in the developing nations of the world as well as in developed countries for fulfilling basic healthcare purposes [21]. It has been estimated by WHO that one-fourth of modern medicines are prepared from plants that had prior traditional uses [22]. Despite living in the present period of synthetic medicine, research-based drug discovery from ethnomedicine is significantly successful to a large extent.

The open field and hole board tests are very convenient and paramount methods for determining the neurological potential of medicinal plant extracts [23]. In these experiments, the presence of agents with sedative properties will reduce movements and interrupt in the interest of the new milieu. The present experimental extract at 250 and 500 mg/kg unveiled a significant inhibition of locomotion which caused a gradual reduction in movement in mice compared to the control (Figure 2). The sedative action was first noticed at 30 min and continued until the last observation period (120 min). The head-dipping behavior of animals in the hole board test is directly related to their emotional state [24]. From our present study, it was evident that the extract had a significant ($p < 0.05$) CNS depressant effect at 250 and 500 mg/kg (Figure 3). The neuropharmacological activity of the extract could be associated with the compounds distinguished by GC-MS profiling. As the major identified bioactive constituent of the extract, 9,12,15-octadecatrienoic acid, (z,z,z), was reported to exert psychotropic effects [25]. Nevertheless, further isolation of bioactive compounds and in-depth study is required to reveal the mechanism of action of these compounds beyond their neurological effects.

Neurons are predominantly susceptible to oxidative damage owing to their high dependence on oxygen consumption, copious polyunsaturated fatty acids in their membrane, and an enfeebled antioxidant defense mechanism [26,27]. Progressively, it damages neuron structure and impairs neuron function, ultimately causing the development of neurodegenerative disorders. ROS-induced oxidative stress is a vital factor in the pathogenesis of AD as it causes the accumulation and deposition of amyloid peptides, hyperphosphorylation of tau protein, modulation of JNK/MAPK stress-activated protein kinase pathways, and oxidation of functional biomolecules [3,28–30]. Therefore, phytochemicals with antioxidant activity, through reduction of oxidative damage, may play a vital role in treating AD. The GCMS identified phytochemicals of the extract 9,12,15-octadecadtrienoic acid (z,z,z) [31]; heptadecanoic acid, ethyl ester [32]; chloroacetic acid, tetradecyl ester [33]; neophytadiene [34] as having antioxidant activity. Therefore, plant-based therapies that target the relationship between oxidative stress and neurodegeneration at the cellular and molecular level may improve treatment and drug development efforts. In this regard, *G. tetrandra* can be a source of pharmacophore (s) for that particular purpose.

Nowadays, computer-aided drug design (CADD) is a buzzword across the world to identify drug candidates from diverse sources [35]. Another tremendous screening approach is ADMET profiling, which analyses the molecular weight, lipophilicity, number of hydrogen acceptors and donors, various types of toxicity, and so on. Pharmacokinetic and pharmacophore analysis give extra advantages in predicting the appropriate drug compounds, and Table 3 shows the properties that are important for drug discovery. All the compounds showed drug-like properties and most of the phytochemicals displayed high intestinal absorption rates, but two were low. Toxicity is harmful to the body and most of the toxicity tests on an animal model are expensive and time-consuming [36]. A considerable range of $LD_{50}$ values were found and that means the toxicity level of the bioactive compounds was low and other toxicity levels like hepatotoxicity and AMES toxicity levels were zero. The blood-brain barrier (BBB) is an important layer that prevents different types of solutes from entering the nervous system. To be a neurological drug, any compound needs to cross BBB [37]. Table 2 showed that all the compounds have the ability to cross the BBB. Furthermore, each and every compound exhibited drug-like characteristics and did not show any Lipinski rule violations.

Molecular docking is a process that reveals the best binding affinity with minimal energy of protein-ligand complexes. In molecular docking studies, the lowest binding energy refers to the best binding affinity of a ligand to a targeted protein [38]. According to the present study, one compound named 1,5-diphenyl-2h-1,2,4-triazoline-3-thione (CID: 2802516) has shown a higher binding affinity ($-5.5$ Kcal/mol) than the control rivastigmine ($-4.2$ Kcal/mol). On the other hand, two ligands, 1-hexanone, 1-phenyl- (CID: 70337), and 3-n-hexylthiolane, s,s-dioxide (CID: 543842) have displayed the same docking score ($-4.2$ Kcal/mol) as the control drug. Lower binding affinity than control has been obtained by the rest of the compounds and all the data is mentioned in Table 4. Hence, chemical analysis, in vivo neurological activity assessment, and in silico studies exposed the evidence of folklore use of *G. tetrandra* in mental disorders.

## 5. Conclusions

The current study has divulged the neurological effect of *G. tetrandra* extract in in vivo open field and hole board tests. Besides, among the GCMS identified compounds, 1,5-diphenyl-2h-1,2,4-triazoline-3-thione (Pub Chem CID: 2802516) showed the better interaction with β-amyloid precursor protein. Furthermore, no toxicity was found for the extract in both the brine shrimp lethality bioassay and in silico study data. Hence, both experimental and computational studies revealed the effectiveness of *G. tetrandra* in neurological disorders, especially in Alzheimer's disease. It is recommended to isolate bioactive phytochemicals and to explicate the mechanistic pathway for preventing neurological disorders.

**Author Contributions:** Conceptualization: M.N.H.Z.; Methodology: M.S.R., M.M.I., F.Y.; Software: M.A.I., M.M.H., P.B.; Formal analysis: M.S.R., M.N.H.Z., M.N.H.; Data curation: M.S.R., M.N.H.Z., M.A.I.; Writing-original draft preparation: M.N.H.Z., M.S.R., M.A.I.; Writing-review and editing: M.N.H., A.H., M.A.R.; Supervision: M.N.H.Z., M.N.H.; Funding acquisition: M.N.H., B.K. All authors have read and agreed to the published version of the manuscript.

**Funding:** This research was supported by Basic Science Research Program through the National Research Foundation of Korea (NRF) funded by the Ministry of Education (NRF-2020R1I1A2066868), the National Research Foundation of Korea (NRF) grant funded by the Korean government (MSIT) (No. 2020R1A5A2019413). The research project was also supported by a grant from the Jashore University of Science and Technology (Grant JUST/Research Cell/FoBST-03-/2020-21).

**Institutional Review Board Statement:** Not applicable.

**Informed Consent Statement:** Not applicable.

**Data Availability Statement:** Not applicable.

**Acknowledgments:** Authors are grateful to the authorities of the department of genetic engineering and biotechnology and the department of pharmacy, Jashore University of Science and Technology for providing excellent working facilities.

**Conflicts of Interest:** The authors declare that they have no competing interests.

**List of Abbreviations:**

AD: Alzheimer's disease; ROS: Reactive Oxygen Species; CREB: cAMP Response Element-Binding Protein; GSK-3: Glycogen Synthase Kinase-3; GAE: Gallic Acid Equivalent; QE: Quercetin Equivalent; GCMS: Gas Chromatography-Mass Spectroscopy; DPPH: 2,2 diphenyl-1-picrylhydrazyle; FRAP: Ferric Reducing Antioxidant Power; b.w: Body Weight; ADME: Absorption, distribution, metabolism, and excretion; APP: Beta-amyloid Precursor Protein; CNS: Central Nervous System; Aβ: Beta-amyloid; JNK/MAPK: c-Jun N-terminal Kinase of Mitogen-activated Protein Kinase family; CADD: Computer-Aided Drug Design; BBB: Blood-Brain Barrier.

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
