# Peer review of "In Vivo Neuropharmacological Potential of Gomphandra tetrandra (Wall.) Sleumer and In-Silico Study against β-Amyloid Precursor Protein"

_processes, doi:10.3390/pr9081449_

Round 1

Reviewer 1 Report

In this paper, the authors describe the isolation and the biological evaluation of various compounds from Gomphandra Tetrandra. While the results are scientifically sound and interesting, I don't see why the authors chose to submit this paper to Processes - publications such as Molecules, Pharmaceutics or IJMS from the same editor would have probably been more appropriate. As it stands, there is nothing particularly novel in the methods the authors chose to apply for this work, and as such I do not consider this article suitable for publication in Processes. I would encourage them, however, to resubmit the paper to a more appropriate publication.

That said, the paper is also somewhat hard to read and it is difficult to see the authors' point in some passages of the introductions and in the conclusion. I recommend extensive English editing and revisions.

Author Response

Point 1: In this paper, the authors describe the isolation and the biological evaluation of various compounds from Gomphandra Tetrandra. While the results are scientifically sound and interesting, I don't see why the authors chose to submit this paper to Processes -publications such as Molecules, Pharmaceutics or IJMS from the same editor would have probably been more appropriate. As it stands, there is nothing particularly novel in the methods the authors chose to apply for this work, and as such I do not consider this article suitable for publication in Processes. I would encourage them, however, to resubmit the
paper to a more appropriate publication.

>> (Response) We are grateful to the reviewer to raise this issue. As there was not much research articles in close relationship of neuropharmacological features of G. tetrandra leaves extract as a natural agent against neurological disorders, especially Alzheimer’s disease. Here, we submitted our work in Special Issue of "Secondary Metabolites: Extraction, Optimization, Identification and Applications in Food, Nutraceutical, and Pharmaceutical Industries" which contain keywords such as Phytochemical characterization, Pharmacological activity, Secondary metabolites, Bioactive compounds, Functional ingredients, and Liquid and gas chromatographies. As, our study is highly fit with this Special Issue of the journal of ‘Processes’, therefore, we submitted this journal. Hope the reviewer will understand our situation.

Point 2: That said, the paper is also somewhat hard to read and it is difficult to see the authors' point in some passages of the introductions and in the conclusion. I recommend extensive
English editing and revisions.

>> Response: We significantly modified our revised manuscript. In the introduction part, we discussed different mechanistic pathways of Alzheimer’s disease and the pathogenesis of the disease. That’s why it may look somewhat hard, but it is quite common and relevant to our study. Besides, the conclusion section has been modified. Furthermore, English writing has been checked again and corrected as possible on pages 1, 2, 3, 4, 5, 6, 7, 8, 9, 10, 12, 13, 14, and 15; and the corrected parts have been marked in green.

Reviewer 2 Report

The authors of the article determine the content of phenolic compounds and flavonoids in the extract. However, it is not clear why they are doing this. The identified substances do not belong to these groups of natural compounds.

The identification of compounds in the extract by GC / MS is questionable. The use of high temperatures (over 200 degrees Celsius) can cause the decomposition of native compounds contained in the extract.

The authors, unfortunately, do not use the extract derivatization method or another analytical method, for example, HPLC / MS.

In our opinion, the data for the identification of compounds are not sufficient and need to be confirmed.

Author Response

Point 1: The authors of the article determine the content of phenolic compounds and flavonoids in the extract. However, it is not clear why they are doing this. The identified substances do not belong to these groups of natural compounds.

Response: We think that phenolic compounds and flavonoids are potent antioxidants. Also, several studies have reported a positive correlation between phenol and flavonoids content and antioxidant activity. Furthermore, free radicals play a critical role in neuronal deficits. Therefore, to determine the strength of antioxidant activity of extract as well as to relate with neurological potential, we have assessed total content of phenol and flavonoids.

Point 2: The identification of compounds in the extract by GC / MS is questionable. The use of high temperatures (over 200 degrees Celsius) can cause the decomposition of native compounds contained in the extract.

Response: For your kind information, we want to inform you that for GC/MS analysis, the use of 200-300oC is very common and it does not breakdown the constituents. This some reference paper: (https://doi.org/10.1038/s41598-020-73442-0; https://doi.org/10.1038/s41598-020-75722-1; https://doi.org/10.1186/s12906-020-2848-2; https://doi.org/10.1186/s43094-021-00208-4).

Point 3: The authors, unfortunately, do not use the extract derivatization method or another analytical method, for example, HPLC/MS.

Response: Due to our lab limitations and facilities, we could not performed to further chemical analysis, i.e., UHPLC/MS/MS, and NMR etc.

Point 4: In our opinion, the data for the identification of compounds are not sufficient and need to be confirmed.

Response: To validate the structure of a chemical molecule, it must be isolated in pure form and then determined using different analytical procedures, such as NMR (Carbon, hydrogen), Mass spectroscopy, and so on. We were unable to establish the chemical structure using advanced analytical methods due to various instrument limitation. However, as part of our investigations, we hope to be able to do so in the future.

Reviewer 3 Report

The present study focused on identification of bioactive phytochemicals in G. tetrandra, in-vivo evaluation of neuropharmacological potential, and in-silico study of beta-amyloid precursor protein, responsible for Alzheimer’s disease. The paper is interesting and gives new information. However, there are some critical points that need to be clarified or supplemented before publication.

  • Organization of the manuscript is wrong. The article is drafted very chaotic. The work abounds in stylistic and spelling errors and typos. The article is written by various fonts.
  • Please provide full latin names of the plants, for example: Bacopa monnieri should be Bacopa monnieri L., Celastrus paniculatus should be Celastrus paniculatus Willd., etc.
  • Conclusions are incomplete and should be supplemented.
  • The Authors should familiarize themselves with the proper format for References and make appropriate corrections.
  • Many items in the literature are out of date. Please, supplement the References with new scientific reports (published during the last three years).

Author Response

Point 1: Organization of the manuscript is wrong. The article is drafted very chaotic. The work abounds in stylistic and spelling errors and typos. The article is written by various fonts.

Response:  All of the abovementioned questions have been addressed in the required position of the manuscript at page numbers 1,2,3,4,5,6,7,8,9,10, 12,13, and 14.

Point 2: Please provide full latin names of the plants, for example: Bacopa monnieri should be Bacopa monnieri L., Celastrus paniculatus should be Celastrus paniculatus Willd., etc.

Response: The botanical names of the plants have been corrected on page number 2.

Point 3: Conclusions are incomplete and should be supplemented.

Response: The conclusion section of the manuscript has been revised at page number 14.

Point 4: The Authors should familiarize themselves with the proper format for References and make appropriate corrections.

Response: All the references have been corrected according to the required format of the journal on pages 18, 19 and 20.

Point 5: Many items in the literature are out of date. Please, supplement the References with new scientific reports (published during the last three years).

Response: The out dated literature (reference number 20, 25, 33, 35, 38) has been supplemented with the latest references on pages 19 and 20.

Round 2

Reviewer 2 Report

The results they describe in the manuscript are preliminary.

This is confirmed by the authors' answer (see response 4).

Unfortunately, the Conclusion section does not contain this conclusion.

Response 2 is inconclusive.

In our opinion, the conclusions of the authors are premature.

Additional experiments are needed.

Author Response

Point1: The results they describe in the manuscript are preliminary.

>> (Response) The present manuscript describes about the chemical identification, different in vitro tests, in vivo activity in animal model in silico study about Alzheimer’s diseases. All these experiments correlate to the background of the study.

Point 2: This is confirmed by the authors' answer (see response 4).

>> (Response) We are grateful to the reviewer.

Point 3: Unfortunately, the Conclusion section does not contain this conclusion.

>> (Response) The conclusion section has been revised at page number 14.

Point 4: Response 2 is inconclusive.

>> (Response) The completely vaporized sample in separating column equilibrates with stationary and mobile phase. Though this condition is compound dependent, it is affected by temperature. Higher the temperature, more the equilibrium shifted towards mobile phase, that means better separation. But, we don’t use isothermal temperature of 250-3000C. We have used programmed temperature (60-2400C) in where 2400C was used for 4 minutes. This programmed temperature was used to prevent recondensation of higher boiling components and also to pass the components sufficiently fast through the column.  Also, to provide good thermal stability with minimal bleed generation and condensation of water in detector are some reasons to use programmed high temperature in GCMS. That’s why compounds of the sample don’t degrades even we use this programmed high temperature.

Point 5: In our opinion, the conclusions of the authors are premature.

>> (Response) The conclusion section has been revised at page number 14.

Point 6: Additional experiments are needed.

>> (Response) It would better if we performed few other extra and advanced experiments. But, due to current pandemic situation, and some unavoidable technical issues it is somewhat difficult for us to forward to new experiments. As the continuation of our research on neurological activity of medicinal plants of Bangladesh, in future, we hope to do that.